green chemistry/synthetic chemistry

hydrogenation, γ-valerolactone, flow chemistry, heterogeneous catalysis

**Author for correspondence:**
László T. Mika
e-mail: laszlo.t.mika@mail.bme.hu

This article has been edited by the Royal Society of Chemistry, including the commissioning, peer review process and editorial aspects up to the point of acceptance.

# Continuous flow hydrogenation of methyl and ethyl levulinate: an alternative route to γ-valerolactone production

József M. Tukacs[1], Áron Sylvester[1], Ildikó Kmecz[1], Richard V. Jones[2], Mihály Óvári[3] and László T. Mika[1]

[1]Department of Chemical and Environmental Process Engineering, Budapest University of Technology and Economics, Műegyetem rkp. 3, Budapest H-1111, Hungary
[2]ThalesNano Nanotechnology Inc, Záhony u. 7, Budapest H-1031, Hungary
[3]MTA Centre for Ecological Research, Institute for Danube Research, Karolina u. 29, Budapest H-1113, Hungary

LTM, 0000-0002-8520-0065

Heterogeneous continuous transformation of methyl levulinate (ML) and ethyl levulinate (EL) to γ-valerolactone (GVL), as a promising $C_5$-platform molecule was studied at 100°C. It was proved that the H-Cube® continuous hydrogenation system equipped with 5% Ru/C CatCart® is suitable for the reduction of both levulinate esters. While excellent conversion rates (greater than 99.9%) of ML and EL could be achieved in water and corresponding alcohols, the selectivities of GVL were primarily affected by the solvent used. In water, 100% conversion and *ca* 50% selectivity that represent *ca* 0.45 $mol_{GVL}\,g_{metal}^{-1}\,h^{-1}$ productivity towards GVL, were obtained under 100 bar of total system pressure. The application of alcohols as a solvent, which maintained high conversion rates up to 1 ml min$^{-1}$ flow rate, resulted in lower productivities (less than 0.2 $mol_{GVL}\,g_{metal}^{-1}\,h^{-1}$) of GVL. Therefore, from a synthesis point of view, the corresponding 4-hydroxyvalerate esters could be obtained even at a higher reaction rate. The addition of sulfonated triphenylphosphine ligand (TPPTS) allowed reduction of the system pressure and resulted in the higher selectivity towards GVL.

## 1. Introduction

The unpredictable reserves of fossil-based resources and increasing efforts to reduce the $CO_2$ emission have directed researchers' attention towards the utilization of alternative

**Figure 1.** Proposed continuous production of GVL from levulinic acid esters.

feedstocks for the chemical industry. In this regard, the transformation of lignocellulose or even valorization of low-cost biomass-based waste streams have had a key position in the fossil-independent value chain of carbon-based consumer products [1,2]. The expanding research activity on biomass conversion has identified several new platform chemicals, which could partially or fully replace the currently used building blocks of well-known synthesis schemes [3–5]. Therefore, numerous strategies have been developed for conversion of carbohydrate and lignin fractions of biomass into platform chemicals [6,7], among which levulinic acid (LA) [6,8], and its derivative γ-valerolactone (GVL) [9] can be distinguished as promising C5-building blocks. Owing to outstanding properties of GVL, a huge variety of its utilization has already been demonstrated including its application as a fuel additive [9,10], a lighter fluid [11] and as a solvent for catalysis [12–17]. Furthermore, it can be used for the production of hydrocarbons [17,18] and fine chemicals [18–21]. Consequently, the production of GVL from lignocellulosic biomass has received emerging interest.

While the reduction of LA has been very extensively studied in the presence of both homogeneous [22,23] and heterogeneous catalysts [24], in comparison, the hydrogenation of its esters even under mild continuous conditions have received limited attention [25]. Because both methyl levulinate (ML) and ethyl levulinate (EL) can be more efficiently prepared by acid-catalysed dehydration of carbohydrates in corresponding alcohols compared with LA [26–28], their utilization as feedstock could open a more efficient route for the production of GVL. Although homogeneous catalysts represent outstanding activity and selectivity [29], to facilitate catalyst recycling, several heterogeneous Ru- [30–32], Co- [33] and Cu-based [34] systems using molecular $H_2$ were tested under batch conditions. Since transfer hydrogenation protocol also offers a safer way to the direct reduction of the $C = O$ group, several systems were reported on GVL production by using Raney-Ni® [35], metal oxides [36–39] and hydroxides [40,41], and metal-organic-framework (MOF) [42] as catalysts and different alcohols as H-donors. Some of these systems were also tested under continuous conditions [43–47]. In general, however, the higher temperatures (130–240°C) were necessary to give reasonable yields in these systems [33,48], and in some cases, a special catalyst such as MOF was necessary to obtain reasonable performance.

For high-throughput heterogeneous catalytic reduction, one of the most promising techniques is the H-Cube® continuous flow hydrogenation system (electronic supplementary material, figure S1), which was developed by ThalesNano Inc. [49]. In this system, hydrogen is generated *in situ* by the electrolysis of water and then continuously mixed with the substrate through a special mixer unit establishing the given total system pressure. This technique representing an excellent example of popular flow chemistry [50,51] avoids handling of high-pressure hydrogen gas from gas cylinders making the process much safer. In addition, it helps to develop a safer and higher-volume process for the reduction of levulinate esters. However, commercially available catalysts such as Ru/C, Pd/C and Pt/C catalysts have not been investigated for the reduction of ML and EL in H-Cube® system.

Herein, we report the evaluation of an environmentally benign and safer alternative protocol of the production of GVL from levulinic acid esters by applying H-Cube® continuous flow hydrogenation equipment (figure 1).

# 2. Materials and methods

## 2.1. Materials

Chemicals (ML, EL and γ-valerolactone) were purchased from Sigma–Aldrich Kft. (Budapest, Hungary) and used as received. Sulfuric, hydrochloric and nitric acids for the preparation of ICP–MS samples were purchased from Merck Hungary Kft. (Budapest, Hungary). TPPTS ((3,3′,3″-phosphanetriyltris-(benzenesulfonic acid) trisodium salt), $P(C_6H_4\text{-}m\text{-}SO_3Na)_3$) was prepared by published method [52].

## 2.2. Experimental set-up and procedure

Catalytic hydrogenation experiments were performed in H-Cube® continuous flow hydrogenation system (ThalesNano Nanotechnology Inc., Budapest, Hungary) equipped with CatCart® (height: 30 mm, Ø 5 mm) tubular reactors containing different catalysts as follows: 5% Ru/C (140 mg, 7 mg Ru) and 10% Pd/C (140 mg, 14 mg Pd). The stock solution of ML or EL was prepared by dissolving 0.260 g (2 mmol) of ML or 0.288 g (2 mmol) of EL in 20 ml of corresponding solvent (water, ethanol or methanol) resulting in the concentration of the substrate of $0.1\ mol\ l^{-1}$. In a typical experiment, the stock solution was loaded into the tubular reactor by an HPLC pump. The reaction temperature was adjusted to 100°C. The reaction mixtures were analysed by GC.

## 2.3. Sample analysis

GC analyses were performed on an HP 5890 instrument with Restek Rtx®-5 capillary column (30 m × 0.25 mm × 0.25 μm) using $H_2$ as a carrier gas. For the analysis, 10 μl of the reaction mixture was dissolved in 1 ml of methylene chloride followed by the addition of 10 μl toluene as an internal standard. The Ru content of the reaction solution was determined by ICP–MS as follows: in an Eppendorf vial, 50 μl of 37% HCl was added to 1.0 ml of reaction sample. Each 0.5 ml of these solutions was pipetted into 15 ml single-use polypropylene centrifuge tubes (MetalFree grade, VWR, Radnor, PA, USA), 100 μl 65% $HNO_3$ and 20 μl of $20\ mg\ l^{-1}$ indium internal standard solution were added, and the solutions were filled up to 5 ml with high purity water. These sample solutions were analysed by SF–ICP–MS Instrument type Element2 (ThermoFinnigan, Bremen, Germany). The operating conditions are summarized in the electronic supplementary material, table S10.

$^1$H NMR spectra were recorded on Bruker Avance 250 MHz spectrometer. Methyl 4-hydroxypentanoate: $^1$H NMR (250 MHz, CDCl$_3$): δ 1.18 (d, 3H), 1.82 (m, 2H), 2.45 (t, 2H), 3.65 (s, 1H), 3.67 (s, 3H), 3.84 (m, 1H). Ethyl 4-hydroxypentanoate: $^1$H NMR (250 MHz, CDCl$_3$): δ 1.18 (d, 3H), 1.29 (t, 3H), 1.82 (m, 2H), 2.45 (t, 2H), 3.65 (s, 1H), 3.84 (m, 1H), 4.03 (m, 2H). γ-valerolactone: $^1$H NMR (250 MHz, CDCl$_3$): δ 1.38 (d, 3H), 1.93–2.38 (m, 2H), 2.27–2.56 (m, 2H), 4.66 (m, 1H)

# 3. Results and discussion

We demonstrated that LA can be converted to GVL in H-Cube® reactor under 100 bar of $H_2$ at 100°C exhibiting productivity (hereafter P ($mol_{GVL}\ g_{metal}^{-1}\ h^{-1}$)) of 0.83 for Ru/C and 0.2 for Pd/C in water at 100 bar, respectively [53]. From the viewpoint of green chemistry, the use of water or alcohols as reaction media are much more favourable, since they have been considered as environmentally benign or even renewable-based solvents having low negative impacts on the environment [54].

Firstly, we compared Ru and Pd catalysts for the reduction of ML and EL in water as well as in methanol (for ML) and ethanol (for EL), which form during the self-esterification of methyl 4-hydroxypentanoate (MHP) or ethyl 4-hydroxypentanoate (EHP). Moreover, from an additional practical point of view, both water [55] and $C_1$–$C_2$ alcohols [56] can easily be separated from GVL by simple vacuum distillation.

Initially a CatCart® was filled with 5% Ru/C catalyst and used for reduction of 0.1 M solutions of corresponding esters with a flow rate of $1\ ml\ min^{-1}$ under 100 bar of total system pressure at 100°C (table 1, entries 1–4), which is the maximum operating temperature of H-Cube®. Obviously, at lower reaction temperatures, lower reaction rates would be provided. While complete conversions (greater than 99.9%) were achieved for ML and EL in both media at 100°C, the selectivities towards GVL were significantly affected by the solvent used. When reactions were performed in water, moderate (ca 51%) selectivities of GVL were detected for both substrates (entries 1 and 3) indicating the low rate of

**Table 1.** Continuous hydrogenation of ML and EL in the presence of different catalyst[a].

| no. | substrate | catalyst | solvent | $X_{substrate}$ (%)[b] | $S_{MHP}$ (%)[c] | $S_{EHP}$ (%)[d] | $S_{GVL}$ (%)[e] | $P_{GVL}$[f] |
|---|---|---|---|---|---|---|---|---|
| 1 | ML | Ru/C | water | >99.9 | 47.7 | — | 52.6 | 0.451 |
| 2 | ML | Ru/C | methanol | >99.9 | 68.4 | — | 31.6 | 0.273 |
| 3 | EL | Ru/C | water | >99.9 | — | 49.3 | 50.6 | 0.434 |
| 4 | EL | Ru/C | ethanol | >99.9 | — | 87.7 | 12.3 | 0.056 |
| 5 | ML | Pd/C | water | <1.0 | n.d. | — | n.d. | n.d. |
| 6 | ML | Pd/C | methanol | 19.6 | >99.9 | — | <0.01 | n.d. |
| 7 | EL | Pd/C | water | <1.0 | — | n.d. | n.d. | n.d. |
| 8 | EL | Pd/C | ethanol | 9.1 | — | >99.9 | <0.01 | n.d. |

[a]Conditions: $T = 100°C$, $p = 100$ bar, $C_{substrate} = 0.1$ M, flow rate $= 1$ ml min$^{-1}$, n.d.: not determined.
[b]Conversion of substrate.
[c]Selectivity $= mol_{MHP} \times (mol_{GVL} + mol_{MHP})^{-1}$.
[d]Selectivity $= mol_{EHP} \times (mol_{GVL} + mol_{EHP})^{-1}$.
[e]Selectivity $= mol_{GVL} \times (mol_{GVL} + mol_{MHP\ or\ EHP})^{-1}$.
[f]Productivity $P = mol_{GVL}\ g_{metal}^{-1}\ h^{-1}$.

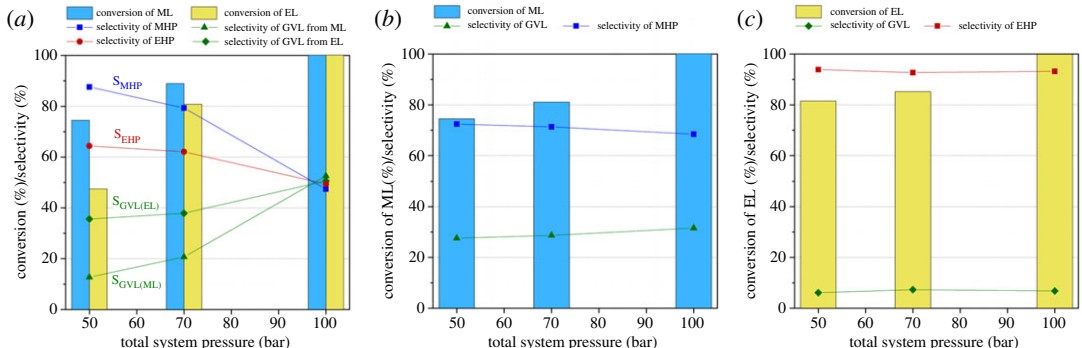

**Figure 2.** Effect of total system pressure on the conversion of ML and EL in water (*a*), ML in MeOH (*b*) and EL in EtOH (*c*). Conditions: $C_{substr} = 0.1$ M, $T = 100°C$, flow rate $= 1$ ml min$^{-1}$. (Detailed data are provided in electronic supplementary material, tables S1–S4.)

de-alcoholization reactions to form GVL at low temperature. It is in a good agreement with previously reported Ru-based systems for hydrogenation of ML in $H_2O$ [32,57]. By using MeOH for conversion of ML and EtOH for EL, the selectivity of GVL decreased by 40% and 75%, respectively (entries 2 and 4) indicating that the self-esterification of 4-hydroxypentanoates to GVL proceeded much slower in alcohols even in the absence of an acidic catalyst at low temperatures [30,37]. It should be noted that through the addition of a catalytic amount of acid into the effluent reaction mixtures at room temperature, no presence of doublet peaks at 1.18 ppm of MHP or EHP was detected by $^1$H NMR any more resulting in the complete formation of GVL with a productivity of 0.857. It corresponds to the value reported for the reduction in LA under similar conditions [53]. In addition, by the one-pot conversion of esters to GVL, the separation of MHP and EHP from neither the corresponding alcohols nor GVL could be considered. It is important to note that neither NMR nor GC-MS analysis shows the formation of other by-product(s) such as 1,4-pentanediol and/or 2-methyltetrahydrofurane, as well as the presence of LA was not detected in the reaction mixture. The application of Pd/C catalyst gave negligible or low conversion rates in all the solvents (table 1 entries 5–8), which corresponds to Rode's observation [30].

It was established that higher yields of GVL were obtained at higher total system pressures [53]. Accordingly, the effect of the total system pressure of the system on the production of GVL from ML and EL was investigated at total system pressures of 50, 70, and 100 bar in water as well as in corresponding alcohols with a flow rate of 1 ml min$^{-1}$ at 100°C (figure 2). As expected, higher conversions could be achieved at higher pressures in all the cases. It is obviously due to the higher concentration of dissolved hydrogen in the liquid phases according to Henry's law [58]. The

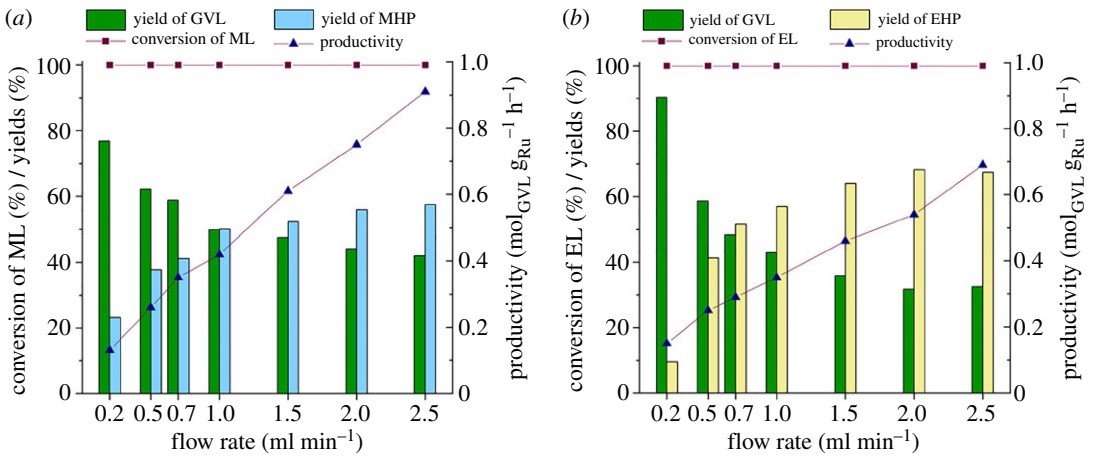

**Figure 3.** The effect of flow rate on the transformation of ML (*a*) and EL (*b*) to GVL in water. Conditions: $C_{substr} = 0.1$ M, $p = 100$ bar, $T = 100°$C. (Detailed data are provided in electronic supplementary material, tables S5 and S6.)

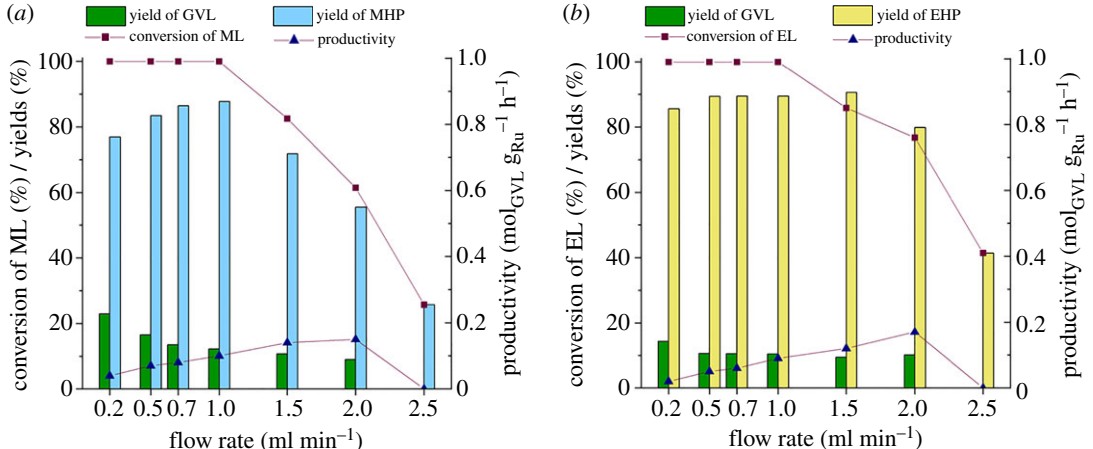

**Figure 4.** The effect of flow rate on the conversion of ML (*a*) and EL (*b*) to GVL in corresponding alcohol. Conditions: $C_{substr} = 0.1$ M, $p = 100$ bar, $T = 100°$C. (Detailed data are provided in electronic supplementary material, tables S7 and S8.)

difference between conversions of ML and EL became negligible when higher total system pressures were applied in water. Moreover, the use of higher total system pressures favoured the GVL formation, which was indicated by its increased selectivity in water (figure 2*a*). The same tendency was reported for reduction of LA [58] and EL [33]. Although, the hydrogenation of the C=O group of both ML and EL could be performed efficiently in the corresponding alcohol at 100 bar, the pressure change had no significant effect on the selectivity of GVL, which remained below 32% and 8% for ML and EL, respectively (figure 2*b,c*). For comparison, we attempted to reduce EL in MeOH, which showed greater than 99% selectivity towards EHP at a flow rate of 1 ml min$^{-1}$ and 100°C, giving a facile continuous method for production of EHP.

The residence time of the reaction mixture could also significantly affect product yield and—therefore—productivity, so we subsequently studied the influence of flow rate with a variation from 0.2 to 2.5 ml min$^{-1}$ on these key production parameters maintaining substrate concentration of 0.1 M, a total system pressure of 100 bar and temperature of 100°C (figures 3 and 4). As shown in figure 3, complete conversion of ML (*a*) and EL (*b*) was measured up to 2.5 ml min$^{-1}$ in water. However, the selectivity, and therefore, the product yield decreased from 76.8% to 42% and 90.4% to 32% for ML and EL, respectively. The higher selectivities of the ML transformation showed that the lactonization of MHP proceeded slightly faster in water than that of EHP under identical conditions. Because no levulinic acid was detected in the reaction mixture, it might be attributed to the faster esterification of MHP to 4-HVA than that of EHP to 4-HVA. A similar tendency was reported for Ru/graphite-catalysed conversion of ML by Shirai and co-workers [32]. Consequently, lower productivities ($P_{max} = 0.69$) were obtained for EL than that of ML ($P_{max} = 0.91$).

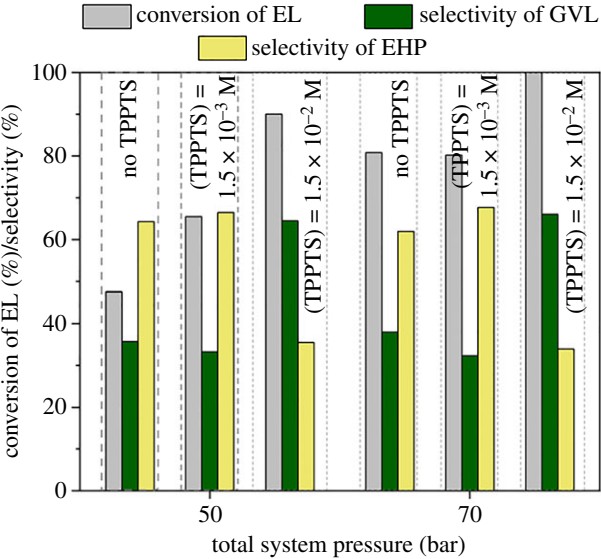

**Figure 5.** Effect of ligand concentration on conversion of EL to GVL in water. Conditions: $C_{substr} = 0.1$ M, flow rate $= 1$ ml min$^{-1}$, $T = 100°$C.

By replacing the reaction medium with the corresponding alcohols (figure 4), conspicuous decreases in conversions were detected over 1 ml min$^{-1}$, and yields of GVL were rather low in both cases. However, the conversions of EL and yields of EHP were slightly higher in EtOH than those achieved for ML and MHP in MeOH. This tendency corresponds to the higher solubility of H$_2$ in higher molecular weight alcohols under identical conditions [59,60]. On the other hand, the alcohol concentration dependence on the equilibrium state of an esterification reaction is well known. This effect was clearly demonstrated by the significantly lower rates of lactone formation from both MHP and EHP in alcohols. Longer residence times favour the lactonization as it was expectedly indicated by slightly higher yields of GVL. It should be noted that the cross-esterification reactions in water, which lead to the formation of 4-hydroxyvaleric acid (4-HVA), cannot be excluded from the reaction sequence. Its transformation to GVL is immediate, the GVL does not react with water at room temperature for three months, and the presence of 4-HVA was only detected at 150°C after a 7-day reaction [61]. This accelerator effect cannot be considered in alcohols, which was indicated by higher concentrations of both MHP and EHP under identical conditions.

It is important to emphasize that the use of the corresponding alcohol as a solvent could allow the synthesis of the corresponding 4-hydroxyvalerate ester with high selectivity, if these are considered as primarily target products.

It was found that sulfonated tertiary phosphine ligands, depending on their concentration, could enhance the activity of the Ru-based hydrogenation catalyst system [62]. It was also proved by applying (C$_4$H$_9$)P(C$_6$H$_4$-m-SO$_3$Na)$_2$ as an additive for aqueous phase conversion of LA to GVL in H-Cube® system [53]. Thus, this influence on GVL production together with the possibility of reduction of total system pressure was investigated via conversion of EL in water (figure 5). In the absence of phosphine, 47.5 and 80.8% conversions were obtained at 50 and 70 bar, respectively (cf. Figure 2a). By applying $1.5 \times 10^{-3}$ M TPPTS ligand, higher conversion (65.5%) could be achieved, which reached 90% in the presence of 10 times higher concentration at 50 bar. At 70 bar, no effect was detected at low ligand feed; however, complete conversion with an increased GVL yields was obtained in the presence of TPPTS with a concentration of $1.5 \times 10^{-2}$ M. The difference between the effect of ligand added and the increased pressure cannot be distinguished. It could be assumed that at higher hydrogen concentration, the influence of such amount of ligand was negligible. Similar findings were reported on conversion of LA, where the increased activity was speculatively attributed to the modified catalyst surface. The interaction of lone electron pair of P-ligand and C=O group was also excluded in our previous study [53]. In order to exclude the possible effect of the dissolved ligand on self-esterification $3 \times 10^{-2}$ mmol of TPPTS was added to the 2 ml of aqueous solution of GVL and EHP with a molar ratio of 1.4. When the reaction mixture was heated up to 100°C for 2 h, no change in the molar composition was shown. Thus, it could be supposed that the modified surface by ligand accelerated both hydrogenation activity and the lactonization reaction. By applying TPPTS, two times

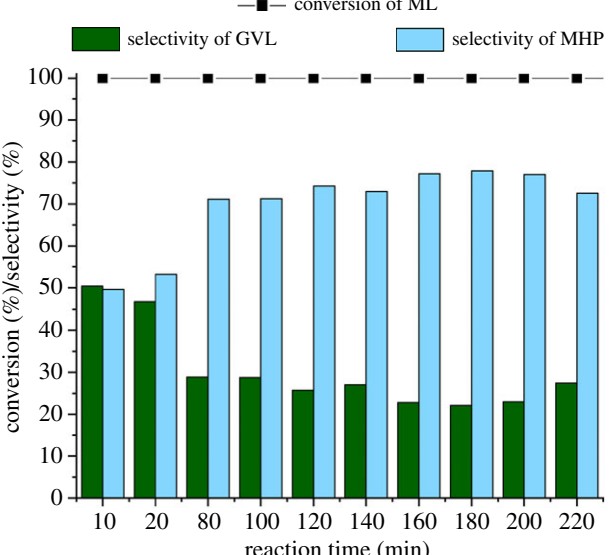

**Figure 6.** Continuous conversion of ML to GVL in water. Conditions: $C_{substr} = 0.1$ M, $p = 100$ bar, $T = 100°$C, flow rate = 1 ml min$^{-1}$.

higher productivity of GVL ($p = 0.567$) was reached, which is *ca* 10 times higher than that reported for transfer hydrogenation system operating even at much higher temperatures [48].

The long-term activity of the catalyst that is a crucial property was tested by using a 0.1 M solution of ML, as for economic aspects, in the absence of TPPTS under 100 bar at 100°C. To conclude, complete conversion was detected for 220 min (figure 6). Although the reason for the unexpected drop of GVL's selectivity between 5 and 40 min is unclear, an average selectivity of $26 \pm 3\%$ was maintained for the next 3 h. In addition, neither ML nor other by-products were detected in the effluent solution. Similar decrease in the selectivity was found for the conversion of ML in iPrOH [47]. In spite of a slightly decreased selectivity, the addition of a catalytic amount of acid to the combined reaction mixture resulted in prompt lactone formation. It was indicated by the disappearance of the peak of MHP at 1.18 ppm (d, 3H).

The Ru-leaching was determined by ICP–MS analysis. The metal leaching after 10 and 20 min reaction time was 3.0 and 2.4 ppb, respectively and became negligible (less than 0.2 ppb) after 80 min (details are given in electronic supplementary material, table S9). Therefore, the possibility of a parallel quasi-homogeneous catalytic transformation that was reported by Köhler *et al.* [63,64] for Pd/C-catalysed Heck reactions could be excluded.

## 4. Conclusion

We demonstrated that the H-Cube® hydrogenation system as an efficient high-throughput flow-chemistry technique can be used successfully for the continuous production of γ-valerolactone from both ML and EL under 100 bar hydrogen at 100°C in water. Complete conversions of both substrates were achieved by using 5% Ru/C CatCart®; however, the selectivity of the transformation was strongly affected by the solvent used. The Ru/C catalyst in the CatCart® remained active even for 220 min. The metal leaching, which could be responsible for parallel quasi-homogeneous catalytic transformations, was negligible. The addition of TPPTS ligand resulted in an enhanced activity that was indicated through a reduction in system pressure.

Data accessibility. All the data are included in the paper as well as further details provided in the article's electronic supplementary material.

Authors' contributions. J.M.T, Á.S. and M.Ó. carried out the experiments and analysed the results. I.K analysed the results and contributed to the interpretation of data. R.V.J. supervised the development of hydrogenation projects in H-Cube systems and prepared the manuscript. L.T.M. supervised the overall direction, design and development of the project and prepared the manuscript.

Competing interests. Authors have no competing interests.

Funding. This work was supported by the National Research, Development and Innovation Office – NKFIH (KH 129508), by the ÚNKP-18-3 New National Excellence Program of the Ministry of Human Capacities, and by the Higher Education Excellence Program of the Ministry of Human Capacities in the frame of Biotechnology research area of Budapest University of Technology and Economics (BME FIKP-BIO). L.T. Mika is grateful for the support of Scholarship of József Varga Foundation, Budapest University of Technology and Economics, Budapest, Hungary.

Acknowledgements. We thank ThalesNano Inc. (Budapest, Hungary) for providing H-Cube® instruments.

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
