## [Reviewer comments · Royal Society Open Science]

Review History

RSOS-182233.R0 (Original submission)

Review form: Reviewer 1 (Adam McCluskey)

Is the manuscript scientifically sound in its present form?

Yes

Are the interpretations and conclusions justified by the results?

Yes

Is the language acceptable?

Yes

Is it clear how to access all supporting data?

No

Do you have any ethical concerns with this paper?

No

Have you any concerns about statistical analyses in this paper?

No

Recommendation?

Accept with minor revision (please list in comments)

Comments to the Author(s)

This was a pleasure to review as the authors have addressed well the initial comments from the three referees from their RSC Advances submission. A few comments though.

Please add the reaction temperature to the paper abstract.

It would have been informative for the authors to have conducted this reaction with the H-CubePro, say at 150°C, which may have been accessible to them as one of the authors (Jones) is cited as at ThalesNano.

It is not clear how the level of leach metals were determined - is this by ICP MS? This should be mentioned in the body text.

Please provide an experimental procedure section.

What is the time (flow rate / volume of reagents) to catalyst exhaustion?

Review form: Reviewer 2

Is the manuscript scientifically sound in its present form?

Yes

Are the interpretations and conclusions justified by the results?

Yes

Is the language acceptable?

Yes

Is it clear how to access all supporting data?

Yes

Do you have any ethical concerns with this paper?

No

Have you any concerns about statistical analyses in this paper?

No

Recommendation?

Reject

Comments to the Author(s)

The authors describe a continuous flow method to screen and optimize heterogeneous catalytic conditions for the conversion of methyl- and ethyl- levulinate esters to gamma-valerolactone

(GVL). Key to this process is the H-Cube hydrogenation system. One benefit of the optimised method is that the H₂ mediated reduction of both levulinate esters proceeds in water and alcoholic solvents (methanol and ethanol). The authors have diligently addressed the reviewer comments from previous iterations of the manuscript, and these efforts have presumably improved the previously submitted versions of the manuscript. However, a number of inadequacies remain:

1. The scalability of the process is not discussed. The practical synthesis of GVL mandates multi-tonne reduction of methyl- and ethyl- levulinate. Did the authors perform gram scale (tens of grams) synthesis to validate the optimized process?
2. The SI is not instructive, and the Tables contain acronyms that are not adequately described. It is therefore difficult to deconvolute the data and the SI is of very limited value in current format.
3. In the SI, Table S1 appears to be directly copied from the marketing brochure and offers no scientific value within the context of the manuscript.

The study does make a minor contribution to the field of continuous flow hydrogenation in describing a method to screen the reduction methyl- and ethyl- levulinate esters to gamma-valerolactone (GVL). However, the H-Cube is a well-established laboratory scale technology and the use of this apparatus for high throughput screening of heterogeneous catalysis (CatCarts) was rapidly established following the inception of the H-Cube device. Of greater concern is the lack of innovative chemistry in the manuscript. For example, the authors had an opportunity to develop a multi-step process to furnish GVL derivatives in a single unit operation or to develop a scalable process for GVL. Given that very narrow focus of the manuscript, I conclude that the manuscript is beyond the scope of the journal. I recommend that the manuscript be considered for publication in a specialised engineering journal to ensure better alignment with the journal scope and readership.

Review form: Reviewer 3 (Huacong Zhou)

Is the manuscript scientifically sound in its present form?

Yes

Are the interpretations and conclusions justified by the results?

Yes

Is the language acceptable?

Yes

Is it clear how to access all supporting data?

Yes

Do you have any ethical concerns with this paper?

No

Have you any concerns about statistical analyses in this paper?

No

Recommendation?

Accept with minor revision (please list in comments)

Comments to the Author(s)

Review comments on RSOS-182233

This resubmitted manuscript has answered most of the issues proposed by previous reviewers, and some revisions were given in the revised manuscript. Although the scientific novelty of this work is indeed not so obvious, the researches of the conversion of levulinic esters to GVL on a continuous reactor were attempted and the results are helpful to promote the real application of the synthesis of GVL through this reactor. So this work is more meaningful in viewpoint of engineering. Therefore, I recommend this work published in Royal Society Open Science after Minor Revision.

1. The author claimed that the addition of acids could promote the formation of GVL in two places in the manuscript but no experiment data were provided. The evidences should be given.
2. A continuous loss of Ru may happen. The authors should detect the Ru content in the catalyst after used for 220 min, not just Ru content in the reaction solution. This may provide more explanations on the loss of activity during long time reaction.
3. The unit in x axe in figure 6 is wrong, min instead of h.

Decision letter (RSOS-182233.R0)

12-Mar-2019

Dear Dr Mika:

Title: Continuous flow hydrogenation of methyl - and ethyl levulinate: an alternative route to gamma-valerolactone production

Manuscript ID: RSOS-182233

Thank you for submitting the above manuscript to Royal Society Open Science. On behalf of the Editors and the Royal Society of Chemistry, I am pleased to inform you that your manuscript will be accepted for publication in Royal Society Open Science subject to minor revision in accordance with the referee suggestions. Please find the reviewers' comments at the end of this email. I apologise that this took longer than usual.

The reviewers and handling editors have recommended publication, but also suggest some minor revisions to your manuscript. Therefore, I invite you to respond to the comments and revise your manuscript.

Please also include the following statements alongside the other end statements. As we cannot publish your manuscript without these end statements included, if you feel that a given heading is not relevant to your paper, please nevertheless include the heading and explicitly state that it is not relevant to your work. We have included a screenshot example of the end statements for reference.

- Ethics statement

Please clarify whether you received ethical approval from a local ethics committee to carry out your study. If so please include details of this, including the name of the committee that gave consent in a Research Ethics section after your main text. Please also clarify whether you received informed consent for the participants to participate in the study and state this in your Research Ethics section.

OR

Please clarify whether you obtained the necessary licences and approvals from your institutional animal ethics committee before conducting your research. Please provide details of these licences and approvals in an Animal Ethics section after your main text.

OR

Please clarify whether you obtained the appropriate permissions and licences to conduct the fieldwork detailed in your study. Please provide details of these in your methods section.

Because the schedule for publication is very tight, it is a condition of publication that you submit the revised version of your manuscript before 21-Mar-2019. Please note that the revision deadline will expire at 00.00am on this date. If you do not think you will be able to meet this date please let me know immediately.

Best wishes,
Dr Laura Smith
Publishing Editor, Journals

On behalf of the Subject Editor Professor Anthony Stace and the Associate Editor Professor John Moses.

RSC Associate Editor:
Comments to the Author:
(There are no comments.)

RSC Subject Editor:
Comments to the Author:
(There are no comments.)

Reviewer comments to Author:
Reviewer: 1

Comments to the Author(s)
This was a pleasure to review as the authors have addressed well the initial comments from the three referees from their RSC Advances submission. A few comments though.

Please add the reaction temperature to the paper abstract.

It would have been informative for the authors to have conducted this reaction with the H-CubePro, say at 150°C, which may have been accessible to them as one of the authors (Jones) is cited as at ThalesNano.

It is not clear how the level of leach metals were determined - is this by ICP MS? This should be mentioned in the body text.

Please provide an experimental procedure section.

What is the time (flow rate / volume of reagents) to catalyst exhaustion?

Reviewer: 2

Comments to the Author(s)

The authors describe a continuous flow method to screen and optimize heterogeneous catalytic conditions for the conversion of methyl- and ethyl- levulinate esters to gamma-valerolactone (GVL). Key to this process is the H-Cube hydrogenation system. One benefit of the optimised method is that the H₂ mediated reduction of both levulinate esters proceeds in water and alcoholic solvents (methanol and ethanol). The authors have diligently addressed the reviewer comments from previous iterations of the manuscript, and these efforts have presumably improved the previously submitted versions of the manuscript. However, a number of inadequacies remain:

1. The scalability of the process is not discussed. The practical synthesis of GVL mandates multi-tonne reduction of methyl- and ethyl- levulinate. Did the authors perform gram scale (tens of grams) synthesis to validate the optimized process?
2. The SI is not instructive, and the Tables contain acronyms that are not adequately described. It is therefore difficult to deconvolute the data and the SI is of very limited value in current format.
3. In the SI, Table S1 appears to be directly copied from the marketing brochure and offers no scientific value within the context of the manuscript.

The study does make a minor contribution to the field of continuous flow hydrogenation in describing a method to screen the reduction methyl- and ethyl- levulinate esters to gamma-valerolactone (GVL). However, the H-Cube is a well-established laboratory scale technology and the use of this apparatus for high throughput screening of heterogeneous catalysis (CatCarts) was rapidly established following the inception of the H-Cube device. Of greater concern is the lack of innovative chemistry in the manuscript. For example, the authors had an opportunity to develop a multi-step process to furnish GVL derivatives in a single unit operation or to develop a scalable process for GVL. Given that very narrow focus of the manuscript, I conclude that the manuscript is beyond the scope of the journal. I recommend that the manuscript be considered for publication in a specialised engineering journal to ensure better alignment with the journal scope and readership.

Reviewer: 3

Comments to the Author(s)

Review comments on RSOS-182233

This resubmitted manuscript has answered most of the issues proposed by previous reviewers, and some revisions were given in the revised manuscript. Although the scientific novelty of this work is indeed not so obvious, the researches of the conversion of levulinic esters to GVL on a continuous reactor were attempted and the results are helpful to promote the real application of the synthesis of GVL through this reactor. So this work is more meaningful in viewpoint of engineering. Therefore, I recommend this work published in Royal Society Open Science after Minor Revision.

1. The author claimed that the addition of acids could promote the formation of GVL in two places in the manuscript but no experiment data were provided. The evidences should be given.
2. A continuous loss of Ru may happen. The authors should detect the Ru content in the catalyst after used for 220 min, not just Ru content in the reaction solution. This may provide more explanations on the loss of activity during long time reaction.
3. The unit in x axe in figure 6 is wrong, min instead of h.

Author's Response to Decision Letter for (RSOS-182233.R0)

See Appendix A.

Decision letter (RSOS-182233.R1)

05-Apr-2019

Dear Dr Mika:

Title: Continuous flow hydrogenation of methyl - and ethyl levulinate: an alternative route to gamma-valerolactone production
Manuscript ID: RSOS-182233.R1

It is a pleasure to accept your manuscript in its current form for publication in Royal Society Open Science. The chemistry content of Royal Society Open Science is published in collaboration with the Royal Society of Chemistry.

On behalf of the Subject Editor Professor Anthony Stace and the Associate Editor Professor John Moses.

RSC Associate Editor
Comments to the Author:
We are satisfied with the revised manuscript.

Reviewer(s)' Comments to Author:

Appendix A

Reviewer: 1

Comments to the Author(s)

This was a pleasure to review as the authors have addressed well the initial comments from the three referees from their RSC Advances submission. A few comments though.

Please add the reaction temperature to the paper abstract.

The reaction temperature was added to the abstract's text.

It would have been informative for the authors to have conducted this reaction with the H-CubePro, say at 150°C, which may have been accessible to them as one of the authors (Jones) is cited as at ThalesNano.

Thank you for this suggestion; however, the H-Cube Pro® was not accessible for these experiments. Obviously, higher applied reaction temperature could result in higher conversion rates at same pressures and/or could allow to reduce total system pressure. It was demonstrated in our previous paper regarding the hydrogenation of levulinic acid (RSC Adv. 2013, 3, 16283.)

It is not clear how the level of leach metals were determined - is this by ICP MS? This should be mentioned in the body text.

The metal leaching was determined by ICP-MS. The body text was extended and the analytical procedures were detailed in the revised section of "Materials and Methods".

Please provide an experimental procedure section.

The section "Materials and Methods" was revised and extended with a detailed experimental and analytical procedure.

What is the time (flow rate / volume of reagents) to catalyst exhaustion?

Checked. The total reaction time for the investigation of catalyst's stability was 220 min. The flow rate (1 mL/min) was indicated in the caption of Figure 6.

Reviewer: 2

Comments to the Author(s)

The authors describe a continuous flow method to screen and optimize heterogeneous catalytic conditions for the conversion of methyl- and ethyl- levulinate esters to gamma-valerolactone (GVL). Key to this process is the H-Cube hydrogenation system. One benefit of the optimised method is that the H₂ mediated reduction of both levulinate esters proceeds in water and alcoholic solvents (methanol and ethanol). The authors have diligently addressed the reviewer comments from previous iterations of the manuscript, and these efforts have presumably improved the previously submitted versions of the manuscript. However, a number of inadequacies remain:

Thank you for your comments.

1. The scalability of the process is not discussed. The practical synthesis of GVL mandates multi-tonne reduction of methyl- and ethyl- levulinate. Did the authors perform gram scale (tens of grams) synthesis to validate the optimized process?

Authors agree that the scalability is very important and crucial issue of a proposed conversion process. However, in the absence of laboratory scale experiments, we cannot evaluate the feasibility of a discussed conversion technology. The basic research activities focusing on continuous biomass valorization processes typically performed on laboratory gram-scale. Please see cited studies. The primarily aim of our work was the evaluation of GVL's production from levulinic acid esters in water or corresponding alcohols under continuous conditions using water as hydrogen source. H-Cube system with higher capacity have developed by ThalesNano and the scale up studies will be subject our further studies.

2. The SI is not instructive, and the Tables contain acronyms that are not adequately described. It is therefore difficult to deconvolute the data and the SI is of very limited value in current format.

3. In the SI, Table S1 appears to be directly copied from the marketing brochure and offers no scientific value within the context of the manuscript.

The SI was revised as follows: 1) The Reviewer is right, the Table S1 contained technical details of H-Cube system, therefore, it was deleted. 2) Abbreviations were added to the SI file for better understanding. 3) Table S10 was revised for better understanding.

The study does make a minor contribution to the field of continuous flow hydrogenation in describing a method to screen the reduction methyl- and ethyl- levulinate esters to gamma-valerolactone (GVL). However, the H-Cube is a well-established laboratory scale technology and the use of this apparatus for high throughput screening of heterogenous catalysis (CatCarts) was rapidly established following the inception of the H-Cube device. Of greater concern is the lack of innovative chemistry in the manuscript. For example, the authors had an opportunity to develop a multi-step process to furnish GVL derivatives in a single unit operation or to develop a scalable process for GVL. Given that very narrow focus of the manuscript, I conclude that the manuscript is beyond the scope of the journal. I recommend that the manuscript be considered for publication in a specialised engineering journal to ensure better alignment with the journal scope and readership.

Reviewer: 3

Comments to the Author(s)

Review comments on RSOS-182233

This resubmitted manuscript has answered most of the issues proposed by previous reviewers, and some revisions were given in the revised manuscript. Although the scientific novelty of this work is indeed not so obvious, the researches of the conversion of levulinic esters to GVL on a continuous reactor were attempted and the results are helpful to promote the real application of the synthesis of GVL through this reactor. So this work is more meaningful in viewpoint of engineering. Therefore, I recommend this work published in Royal Society Open Science after Minor Revision.

Thank you for your kind recommendation.

1. The author claimed that the addition of acids could promote the formation of GVL in

two places in the manuscript but no experiment data were provided. The evidences should be given.

The conversion of MHP to GVL by the addition of HCl was established by GC-MS and NMR technique. The corresponding sentences were modified and the ¹H NMR of MHP and EHP were added to the experimental part for better understanding.

2. A continuous loss of Ru may happen. The authors should detect the Ru content in the catalyst after used for 220 min, not just Ru content in the reaction solution. This may provide more explanations on the loss of activity during long time reaction.

Thank you for your comment. I understand that the determination of the ruthenium content of the catalyst provide information for mass balance of Ru. While metal leaching could be detected in the first two samples, the Ru loss became negligible (<0.2 ppb) after 20 min reaction time. Authors would like to note that no any decrease in the activity was detected for 220 min reaction time (figure 6). The conversion values of methyl levulinate were >99.9%, therefore the negligible change of Ru content had no effect on the catalyst activity.

On the other hand the CatCart is a commercially available product of ThalesNano Nanotechnology Inc. For comparable measurements the Ru content would have been determined before the reaction and after the reaction. Because the non-used catalyst is not available (no samples were taken before reaction), the measurements cannot be compared.

3. The unit in x axe in figure 6 is wrong, min instead of h. The title of x axis of Figure 6 was corrected.